# Estimation of an Image Biomarker for Distant Recurrence Prediction in NSCLC Using Proliferation-Related Genes

**DOI:** 10.3390/ijms24032794

**Published:** 2023-02-01

**Authors:** Hye Min Ju, Byung-Chul Kim, Ilhan Lim, Byung Hyun Byun, Sang-Keun Woo

**Affiliations:** 1Radiological and Medico-Oncological Sciences, University of Science and Technology, Daejeon 34113, Republic of Korea; 2Division of RI-Convergence Research, Korea Institute of Radiological and Medical Sciences, Seoul 07812, Republic of Korea; 3Department of Nuclear Medicine, Korea Institute of Radiological and Medical Sciences, Seoul 07812, Republic of Korea

**Keywords:** non-small cell lung cancer, distant recurrence, RNA-sequence, ^18^F-FDG PET, prediction model

## Abstract

This study aimed to identify a distant-recurrence image biomarker in NSCLC by investigating correlations between heterogeneity functional gene expression and fluorine-18-2-fluoro-2-deoxy-D-glucose positron emission tomography (^18^F-FDG PET) image features of NSCLC patients. RNA-sequencing data and ^18^F-FDG PET images of 53 patients with NSCLC (19 with distant recurrence and 34 without recurrence) from The Cancer Imaging Archive and The Cancer Genome Atlas Program databases were used in a combined analysis. Weighted correlation network analysis was performed to identify gene groups related to distant recurrence. Genes were selected for functions related to distant recurrence. In total, 47 image features were extracted from PET images as radiomics. The relationship between gene expression and image features was estimated using a hypergeometric distribution test with the Pearson correlation method. The distant recurrence prediction model was validated by a random forest (RF) algorithm using image texture features and related gene expression. In total, 37 gene modules were identified by gene-expression pattern with weighted gene co-expression network analysis. The gene modules with the highest significance were selected (*p*-value < 0.05). Nine genes with high protein–protein interaction and area under the curve (AUC) were identified as hub genes involved in the proliferation function, which plays an important role in distant recurrence of cancer. Four image features (GLRLM_SRHGE, GLRLM_HGRE, SUVmean, and GLZLM_GLNU) and six genes were identified to be correlated (*p*-value < 0.1). AUCs (accuracy: 0.59, AUC: 0.729) from the 47 image texture features and AUCs (accuracy: 0.767, AUC: 0.808) from hub genes were calculated using the RF algorithm. AUCs (accuracy: 0.783, AUC: 0.912) from the four image texture features and six correlated genes and AUCs (accuracy: 0.738, AUC: 0.779) from only the four image texture features were calculated using the RF algorithm. The four image texture features validated by heterogeneity group gene expression were found to be related to cancer heterogeneity. The identification of these image texture features demonstrated that advanced prediction of NSCLC distant recurrence is possible using the image biomarker.

## 1. Introduction

Non-small cell lung cancer (NSCLC) has high incidence among cancers and large molecular heterogeneity in tissues [1,2]. Its molecular heterogeneity was shown to be different not only between patients but also between intratumor and intertumor regions [3]. Intratumor heterogeneity is known to be linked to the development of primary tumors and recurrence [4]. It is possible to diagnose cancer by analyzing intracellular gene expression events and find a suitable treatment method for each cancer [5]. Many studies have been conducted to search for methods to diagnose cancers having different genotypes and to find a treatment for each cancer. Image features that analyze phenotypes based on genotype, next-generation sequencing (NGS) for large-scale gene analysis, and radiogenomics combining fluorine-18-2-fluoro-2-deoxy-D-glucose positron emission tomography (^18^F-FDG PET) image features and NGS are a few examples.

Heterogeneity functional gene expression was identified by NGS analysis of NSCLC. NGS is a high-throughput sequencing analysis method that is more capable of accurately quantifying large amounts of gene information than conventional gene analysis methods [6]. The classical imaging techniques use radiation to image the affected area without causing pain to the patient, grasping the overall characteristics of the affected area; it has the advantage of quick analysis [7] but only shows the cancer phenotype. Radiogenomics is a technique that combines image feature technology for analyzing images and NGS technology for mass analysis of genes, revealing the relationship between the expression of specific genes related to cancer and the image features present. By combining the two analysis methods, non-invasive diagnosis and prediction of cancer are possible [8].

^18^F-FDG PET/computed tomography (CT) has the advantage of evaluating metabolic processes in cancer. ^18^F-FDG is absorbed during glucose metabolism, and it is possible to estimate glucose metabolism by imaging the FDG remaining in the cell. Each image feature represents the cellular phenotype and can be used for diagnosis. ^18^F-FDG PET/CT image radiomics analysis was used to analyze lung cancer heterogeneity by Buvat et al. [9]. Studies by Yoon et al. and Bianconi et al. showed that it is possible to understand the current cancer status through image texture analysis of CT images of lung cancer [10,11]. Patients were classified by the tumor, node, and recurrence (TMN) stage for analysis, and recurrence and survival were predicted using ^18^F-FDG PET/CT image analysis by Andersen et al. and Nakajo et al. [12,13]. Ravanelli et al. predicted survival rates through analysis of CT imaging factors associated with mutations in epidermal growth factor receptor (EGFR) [14]. Piñeiro-Fiel et al. showed that image texture features are related to biological heterogeneity in lung cancer [15].

Predictions of distant recurrence have a considerable impact on the survival of patients with NSCLC. In this study, we analyzed the correlation between the expression of proliferation-related genes involved in distant recurrence of NSCLC and quantitative ^18^F-FDG PET image texture features to evaluate an image biomarker for the prediction of NSCLC distant recurrence. The NSCLC distant recurrence prediction model was based on image texture features that are related to gene expression. A radiogenomics-based prediction model that incorporates both imaging features and gene expression can accurately predict the distant recurrence of NSCLC.

## 2. Results

In this study, ^18^F-FDG PET data and RNA-sequencing data from 53 patients with NSCLC were used for analysis. The average age of the patients was 67.5 years, and the ratio of men to women was approximately 8:2 (Table 1). The process of development of the relationship between the RNA-sequencing data and ^18^F-FDG PET image features is shown as a schematic in Figure 1.

### 2.1. Gene Modulation and Hub Gene Assay

To identify hub genes that might play an important role in distant recurrence, weighted gene co-expression network analysis (WGCNA) was used to construct gene modules with similar expression patterns, followed by a network analysis. In total, 37 gene modules were obtained (Figure 2). The violet, dark gray, and cyan modules with highly significant correlations in the distant recurrence group were selected (*p*-value < 0.05). The STRING network was constructed to identify strong PPI genes in each module (node degree ≥ 10). GO term analysis was performed to confirm module functions. Nine genes (RAD54L, NDC80, POLE, BLM, MYBL2, NCAPG2, RECQL4, E2F1, and INCENP) were selected as proliferation-related hub genes (*p*-value < 0.05).

### 2.2. Hub Gene and Image Feature Associations

The analysis was performed using 47 image texture features and nine hub genes. Results regarding the relationship between gene expression levels and texture image features are presented in Figure 3. Four image features (gray-level run length matrix (GLRLM)_HGRE, standard uptake value (SUV)mean, and GLZLM_GLNU) were correlated with the expression of six genes (RAD54L, POLE, BLM, MYBL2, E2F1, INCENP) (*p*-value < 0.1). These genes were enriched in “Mitotic cell cycle” (MYBL2, POLE), “DNA replication” (BLM, POLE), “G1/S transition of mitotic cell cycle” (E2F1, POLE), “DNA duplex unwinding” (BLM, RAD54L), and “Positive regulation of mitotic cell cycle spindle assembly checkpoint” (INCEP), which are functions in the proliferation process.

### 2.3. Creation of the Prediction Model

Gene expression levels and features extracted from PET/CT images were used to create a random forest (RF) model to predict the distant recurrence of NSCLC. The RF model precision, recall, area under the curve (AUC), and accuracy scores with the 47 image texture features identified by functional analysis were 0.692, 0.733, 0.729, and 0.59, respectively. The RF model precision, recall, AUC, and accuracy scores with the nine hub genes identified to be related to proliferation were 0.8, 0.783, 0.808, and 0.767, respectively. The RF model precision, recall, AUC, and accuracy scores of the four image texture features and six genes identified to be correlated were 0.802, 0.792, 0.912, and 0.783, respectively. The RF model precision, recall, AUC, and accuracy scores of the four image texture features with a high probability of significant correlation with hub genes were 0.832, 0.75, 0.779, and 0.738, respectively (Table 2). The RF model test accuracy and AUC using 47 image texture features were 0.727 and 0.732, respectively. The RF model test accuracy and AUC with the nine hub genes were 0.727 and 0.679, respectively. The RF model test accuracy and AUC using the four image texture features and six genes identified to be correlated were 0.818 and 0.804, respectively. The RF model test accuracy and AUC with the four image texture features were 0.727 and 0.732, respectively.

## 3. Discussion

Liao et al. (2020) identified the significance of the mRNA expression-based stemness index (mRNAsi) in the clinical characteristics of LUAD. They found that hub genes highly correlated with mRNAsi by WGCNA analysis. Specifically, they identified RAD54L as one of the hub genes that may have a strong influence on LUAD stem cell maintenance. Our current study aimed to find hub genes that correlate with the distant recurrence of NSCLC using the WGCNA approach. We also suggested that RAD54L could influence the distant recurrence of NSCLC. Bai et al. (2018) confirmed experimentally in mice that glioblastoma cells exhibit radioresistance by upregulating the expression of RAD54L. Shen et al. (2021) identified hub genes to be significantly correlated with the recurrence of LUAD in patients and suggested the possibility of treatment with drugs targeting these hub genes. They used the Kaplan–Meier method to identify eight hub genes that had influence on the survival of LUAD patients and constructed drug–hub gene interaction. In this study, we validated the hub genes with the AUC value and specified image texture features for predicting distant recurrence in NSCLC patients. Anusewicz et al. (2020) revealed differential gene expression regulation through the Notch, Hedgehog, Wnt, and ErbB signaling pathways in lung squamous cell carcinoma (LUSC) and LUAD. They investigated differences in the profiles of downstream target genes in LUSC and LUAD and specified the pathways affecting the progression of cancer before performing gene analysis. They identified E2F1, which is involved in hyperphosphorylation in NSCLC, to significantly alter the prognosis of LUSC patients [16]. In this study, we focused on the differences in gene expression between patients with NSCLC distant recurrence and non-recurrence patients. Consistently, we also identified E2F1 as a hub gene in distant recurrence of NSCLC.

We estimated the heterogeneity image biomarkers of NSCLC that can be used to develop distant recurrence prediction models using heterogeneity-related functional genes. Heterogeneity analysis studies through genetic analysis or image analysis have been performed [17,18], but their respective drawbacks have limited their use [19]. A more accurate heterogeneity diagnosis is possible by fusing two different types of data using radiogenomic analysis [20]. The correlations between the nine proliferation-related hub genes and 47 image texture features identified four image texture features (GLRLM_SRHGE, GLRLM_HGRE, SUVmean, GLZLM_GLNU) that were related to six genes, which could further be used as a predictive image biomarker for cancer heterogeneity. For validation, the accuracy of the machine learning predictive model was evaluated using RNA-sequencing results and image texture features; when 47 image features were used, the accuracy was 0.59, and when the four image texture features and six genes identified by the Pearson correlation method were used, accuracy was 0.783.

Cancer heterogeneity is a phenomenon that occurs due to cell distribution changes in cancer tissue [21]. Cancer cells have functions similar to stem cells, allowing them to self-renew, migrate, and differentiate [22]. With changes in the distribution of cells in the tissue, their heterogeneity increases when these functions are activated. The effects of increased heterogeneity in cancer are as follows: first, cancer treatment becomes more difficult [23]. The effectiveness of treatment with a single anticancer agent is reduced, and there is an increased risk of needing multiple anticancer agents, because as the cancer heterogeneity increases, cancer cells with different properties make up the cancer tissue. Second, migration of cancer cells to secondary sites and external organs occurs, and these cells proliferate, resulting in enlarged cancer tissue, thereby making treatment difficult [24].

Cancer aggressiveness is associated with cell proliferation [25]. In normal tissue, the mitotic cell cycle is controlled; however, cancer tissues sustain proliferative signaling and increases in the cell-cycle pathway [26]. The proliferation of cancer cells leads to the accumulation of genetic abnormalities and heterogeneous changes in the tumor microenvironment [27]. Therefore, by analyzing the expression of proliferation-associated genes, instead of the expression of all genes, one can analyze heterogeneity more accurately. Nine NSCLC proliferation-related hub genes were identified and used for analysis. These proliferation-related genes were further used to evaluate the degree of association between cellular heterogeneity and image biomarkers.

As described above, the current status can be diagnosed through image analysis of lung cancer. This can also be used to predict mortality or cancer recurrence. CT image analysis can be used to estimate recurrence with the tumor size. Analysis with ^18^F-FDG PET/CT imaging can also be useful for determining tumor heterogeneity. The information contained in an image is limited, and it is very difficult to explain the behavior of cancer cells using this alone. To overcome this limitation, we researched genes expressed in relation to imaging factors and confirmed the imaging biomarkers using a combination of gene expression and image texture features. PET image biomarkers for NSCLC recurrence prediction were characterized by combining ^18^F-FDG PET/CT image texture features and genetic analysis.

In this study, four image texture features were confirmed to be sufficient to predict distant recurrence. In general, features such as the maximum SUV (SUVmax), peak SUV (SUVpeak), total lesion glycolysis (TLG), and Entropy_log10 have been previously used for radiogenomic analysis for cancer prediction or cancer recurrence prediction [28]. However, in this study, the correlations (*p*-values) with SUVmax, SUVpeak, TLG, and Entropy_log10 were lower than those for the four image texture features. This result suggests that new factors can be used to develop a model for the improved prediction of recurrence in NSCLC using radiogenomics.

This study demonstrates how to specify image biomarkers through validation of functional gene groups. Because the image characteristics obtained by image analysis are phenotypic, there is a problem regarding reduced accuracy of diagnosis. To solve this problem, we developed a method to increase the prediction accuracy of image features by coupling them with the expression of a set of genes that can represent the genotype. Proliferation-related functional hub genes involved in recurrence and heterogeneity were extracted, and image biomarkers were specified by comparing their relevance with image features. To verify this, the accuracy was estimated using a machine learning method. Using this method, it is expected that phenotype and genotype can be diagnosed simultaneously with high accuracy using imaging biomarkers.

Further research will be required on radiogenomic analysis methods that simultaneously analyze the imaging features and genes of NSCLC and image biomarkers that can diagnose cancer types and behaviors by simultaneously analyzing different types of cancers. Although this study was able to identify imaging biomarkers related to recurrence in a small number of non-small cell patients, it lacks validity for clinical application. Currently, research on image biomarkers using radiogenomic methods is in its infancy, and research has not been conducted on most cancers. More research is needed to properly apply image biomarkers in clinical practice.

## 4. Materials and Methods

### 4.1. NSCLC NGS Data Processing

RNA-sequencing data, patient clinical data, and ^18^F-FDG PET images were acquired from The Cancer Imaging Archive/The Cancer Genome Atlas database (NGS data accession number: GSE103584, PET image data: http://doi.org/10.7937/K9/TCIA.2017.7hs46erv) (Accessed on 1 October 2022). Tissues for RNA-sequencing were whole surgical specimens that were obtained during surgery before treatment. The inclusion criteria were M0 stage patients with recurrence data. Data for 53 patients were classified in a binary manner between distant recurrence (*n* = 19) and non-recurrence (*n* = 34) groups based on clinical data and ^18^F-FDG PET images. Patient information is summarized in Table 1. Acquired data were normalized by Fragments Per Kilobase of transcript per Million (FPKM). The genes with zero FPKM values from all of the samples were trimmed for fast analysis [16].

### 4.2. Weighted Gene Co-Expression Networks and Modules Associated with Clinical Traits

To analyze the correlation between expressed genes and features extracted from images, gene selection was performed. The "goodSampleGenes" function was used to filter the genes with missing entries and zero variance before WGCNA (Version: 1.71). The number of genes used for gene co-expression analysis was 13,060 out of 22,126. To obtain the gene module with the greatest influence for determining distant recurrence, WGCNA was performed [29]. The genes were separated into several modules using the WGCNA tool in the R package. A soft threshold for network construction was selected for gene clustering. In the soft threshold, the adjacency matrix forms a continuous range of values between 0 and 1. The constructed network conforms to the power-law distribution and is close to a real biological network state. A scale-free network was constructed using the blockwise module function, followed by module partition analysis to identify gene co-expression modules, which grouped genes with similar expression patterns. The 37 modules were defined by cutting the clustering tree into branches using a dynamic tree cutting algorithm and assigned different colors for visualization [30]. For each module, the correlation coefficient and statistical significance of the correlation with distant recurrence in patients were calculated. The violet, dark gray, and cyan modules were selected with a *p*-value < 0.05.

### 4.3. Hub Gene Analysis

PPI networks were constructed with the genes of each module using the STRING database (Version: 11.5). The node degree of each gene was calculated to measure the number of interactions in the network. Genes with a node degree >10 were selected with the minimum required interaction score set to 0.9. To confirm the proliferation-related genes in the modules, GO term analysis was performed. The genes enriched in the proliferation-related term with *p* < 0.05 were extracted. AUC value for each gene was calculated. Proliferation-related genes with high AUC values (AUC > 0.6) were selected.

### 4.4. ^18^F-FDG PET Imaging

Tumor volumes were segmented, and radiomics features in the defined tumors were subsequently extracted using the Local Image Features Extraction (LIFEx) version 4.0 software package [31]. The tumor region of interest was drawn using a semi-automated segmentation method with a threshold SUV of 2.0, based on our previous report [32], in three-dimensional images. In the tumor region, the maximum SUV (SUVmax), SUVmean, SUVpeak, metabolic tumor volume (MTV), TLG, and the shape of the histogram were calculated as the intensity features. Gray-level texture features were assessed using four texture matrices: gray-level co-occurrence matrix (GLCM), GLRLM, gray-level size zone length matrix (GLSZM), and neighboring gray-level dependence matrix (NGLDM). The GLCM was calculated in 13 directions with one voxel distance.

The relationship between neighboring voxels and each texture feature calculated from this matrix was the average of the features over the 26 connectivity in space (X, Y, Z). The GLRLM was calculated by number of runs with gray level and length, whereas the GLSZM was computed with the 8-connected region in 2D. The NGLDM was computed from the difference in gray levels between one voxel and all neighbors, and each texture feature was calculated from this matrix [33]. The 47 total image texture features were extracted from the PET image data. The index of extracted image texture features is in Table 3.

### 4.5. Hub Gene and Image Feature Correlation

In total, 47 image features and hub genes were used to estimate the relationship between all factors, which was calculated with the Pearson correlation method and *p*-value using R corrplot package. The image features and genes for inclusion in the distant recurrence prediction model were selected by the *p*-value of the correlation calculated by the t-test (*p*-value < 0.1).

### 4.6. RF Prediction Model Construction

To predict patient outcomes in terms of distant recurrence, we used a machine learning approach [34] called RF [35]. The machine learning prediction model was evaluated quantitatively in terms of the precision, recall, AUC, and accuracy score. To address the data imbalance, the oversampling method was implemented using the Synthetic Minority Over-sampling Technique (SMOTE). The model was evaluated by k-fold validation (K = 10). A total image texture feature (47 image texture features extracted from PET image), a hub gene (nine genes identified as proliferation-related), an image texture feature, a gene (four image texture features and six genes identified as correlated), and an image texture feature (four image texture features with a high probability of significant correlation with hub genes) were used separately for the machine learning using the RF. The RF model was trained with 80% of the 52 patients and tested with 20% of the patients, and the whole dataset was split in a stratified manner.

## 5. Conclusions

Image biomarkers involved in NSCLC recurrence were identified upon validation with proliferation-related genes and were evaluated on the basis of the accuracy of the machine learning prediction model. Through this process, combined analysis of image biomarkers and proliferation-related genes was concluded to be suitable for evaluation of the heterogeneity of NSCLC (78.3% accuracy), compared with the analysis of total image texture features (0.59 accuracy). With an increasing amount of data accumulating from ongoing studies on other cancers, imaging biomarkers together with proliferation-related genes have the potential to be used in clinical applications.

## Figures and Tables

**Figure 1 ijms-24-02794-f001:**
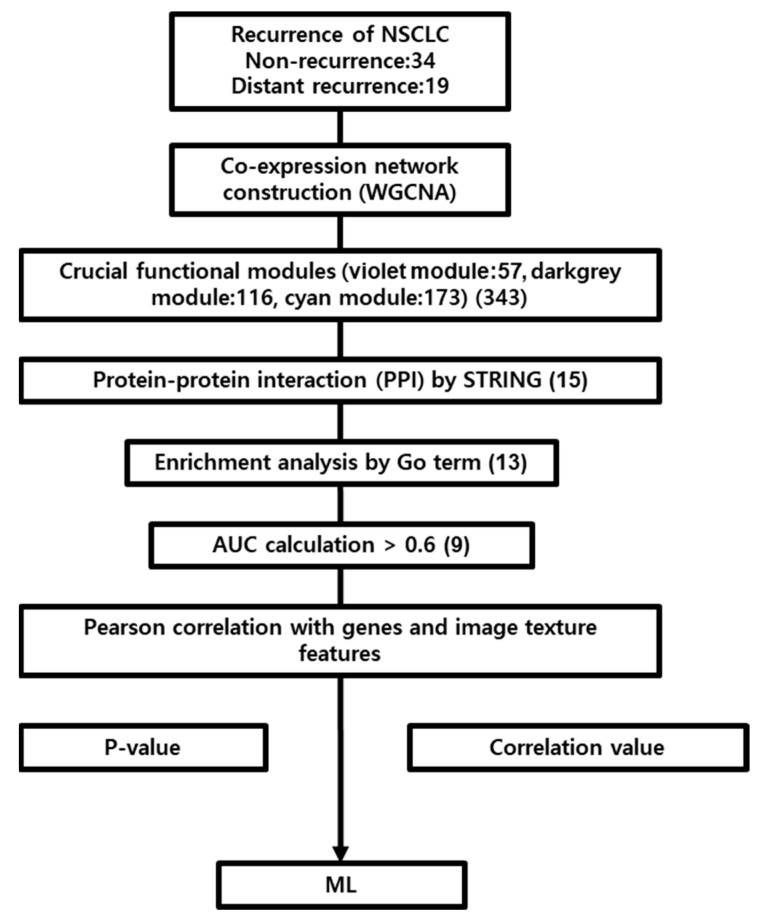
Schematic outline of the study. RNA sequencing data of NSCLC patients with GSE103584 were analyzed by the WGCNA method, and modules with high significant correlation with distant recurrence were identified. The hub genes were intersected for PPI network construction, enrichment analysis, and AUC calculation. The hub genes and image texture features were identified using the Pearson correlation method. The 6 genes and 4 image texture features were significantly correlated. A random forest model of NSCLC distant recurrence was constructed with identified hub genes and image texture features.

**Figure 2 ijms-24-02794-f002:**
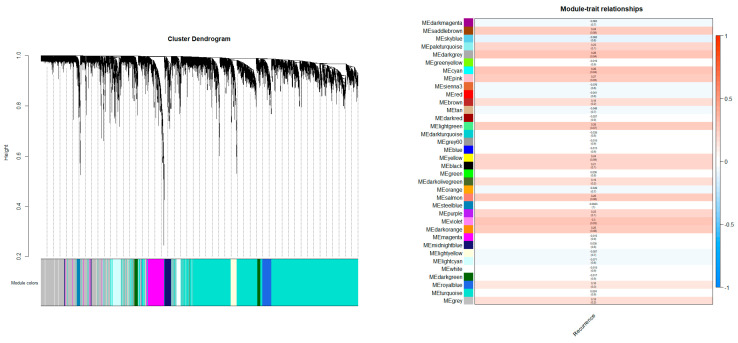
Gene regulation was performed through clustering. In total, 37 gene modules were generated, with each module comprising genes with similar expression patterns (**left**); the relationship between each module and distant recurrence is shown as a heatmap (**right**). The modules relevant to distant recurrence are the violet (module meaning = 0.3, *p*-value = 0.033), dark gray (module meaning = 0.28, *p*-value = 0.041), and cyan (module meaning = 0.28, *p*-value = 0.042) modules.

**Figure 3 ijms-24-02794-f003:**
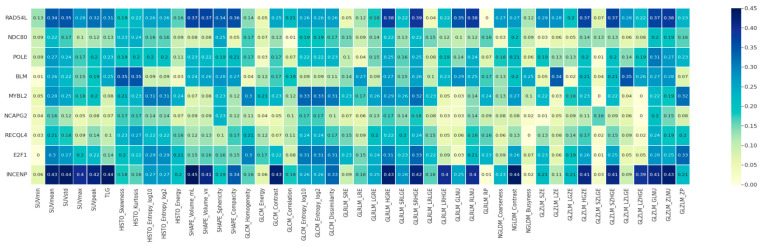
Correlation map of image texture features and functional genes revealing significant associations between 47 image texture features from PET and the 9 hub genes identified by functional analysis.

**Table 1 ijms-24-02794-t001:** List of clinical data for NSCLC patients. The patient age, sex, type of cancer, smoking status, epidermal growth factor receptor (EGFR) mutation, Kirsten rat sarcoma virus (KRAS) mutation, and cancer progression are shown.

Characteristic	Result	Rate
**Average age**	69.28	
**Sex**		
Male	44	83
Female	9	17
**Histology**		
Adenocarcinoma	36	68
Non-small cell lung cancer (not otherwise specified)	2	4
Squamous cell carcinoma	15	28
**Smoking status**		
Current	8	15
Former	37	70
Non-smoker	8	15
***EGFR* mutation**		
Wild	38	72
Mutation	6	11
Unknown	9	17
***KRAS* mutation**		
Wild	32	60
Mutation	12	23
Unknown	9	17
**T stage**		
T1a	10	19
T1b	13	25
T2a	17	32
T2b	5	9
T3	5	9
T4	2	4
Tis	1	2
**N stage**		
N0	41	77
N1	5	9
N2	7	13
**Recurrence**		
No recurrence	34	64
Distant recurrence	19	36

**Table 2 ijms-24-02794-t002:** Precision, recall, AUC, and accuracy values of predictive models created using proliferation-related hub genes and image texture features expressed using the random forest algorithm.

Random Forest	Image Texture Features	Hub Genes	Correlation of Genes and Image Texture	Four Image Texture Features
Precision	0.692	0.8	0.802	0.832
Recall	0.733	0.783	0.792	0.75
AUC	0.729	0.808	0.912	0.779
Accuracy	0.59	0.767	0.783	0.738

AUC, area under the curve; Image texture features, 47 image texture features extracted from PET images; Hub genes, 9 genes identified as proliferation-related; Correlation of genes and image texture, 4 image texture features and 6 genes identified to be correlated; Four image texture features, 4 image texture features that are significantly correlated with hub genes.

**Table 3 ijms-24-02794-t003:** Index of textural features in global, local, and regional areas.

Feature Family	Features
Intensity histogram	Maximum standard uptake value (SUVmax)
	Mean standard uptake value (SUVmean)
	Standard deviation (SUV_SD)
	Total lesion glycolysis (TLG)
	Metabolic tumor volume (MTV)
	1st entropy
Gray-level co-occurrence matrix (GLCM)	Energy
	Contrast
	Entropy
	Homogeneity
	Dissimilarity
Neighboring gray-level dependence matrix (NGLDM)	Contrast
	Coarseness
	Busyness
	Small number emphasis (SNE)
Gray-level run length matrix (GLRLM)	Short run emphasis (SRE)
	Long run emphasis (LRE)
	Gray-level non-uniformity (GLNU)
	Run length non-uniformity (RLNU)
	Low gray-level run emphasis (SRLGE)
	High gray-level run emphasis (SGHGE)
Gray-level size zone matrix (GLSZM)	Small zone emphasis (SAE)
	Large zone emphasis (LAE)
	Gray-level non-uniformity (GLN)
	Zone size non-uniformity (SZN)
	Low gray-level zone emphasis (LGLZE)
	High gray level zone emphasis (HGLZE)

## Data Availability

Not applicable.

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
