# Peer review of "Estimation of an Image Biomarker for Distant Recurrence Prediction in NSCLC Using Proliferation-Related Genes"

_ijms, 2023, doi:10.3390/ijms24032794_

Round 1

Reviewer 1 Report

Ju et al produced a work using TCGA data, both expression and imaging data to identify potential biomarkers using Weighted Gene Co-Expression Networks (WGCNA) in lung cancer associated with recurrence. However some other published articles already worked with this data, Shen et al 2021 (10.3389/fgene.2021.756235), Liao et al 2020 (10.3389/fgene.2020.0031), Anusewicz et al 2020 (https://www.nature.com/articles/s41598-020-77284-8) for example. What this present work brings different and how it is compared to these previous published articles? It needs to bring this similarities and differences in the results and discussion.

Reviewer 2 Report

The manuscript entitled "Estimation of an image biomarker for distant recurrence prediction in NSCLC using proliferation-related genes" by Min Ju et al. aims to identify biomarkers for distant-recurrence in NSCLC combining NGS and PET-imaging. The analysis was then performed using WGCNA, network analysis and correlation analysis. Finally a random forest model was built.

The study certainly presents a clinical relevance and appear to sound.

However, there are major comments that would need to be addressed:

General:

Figures quality needs to be improved and evaluation and test for random forest model are missing.

Material and Methods:

- The NGS data was filtered before applying WGCNA and I completely agree with that. However, I believe that a more stringent filtering would have worked better since WGCNA is based on correlation between genes. How much is the intra-correlation of each module?

-In the hub gene analysis how you decided to use >10 as threshold for selecting hub genes?

-Would be interesting to see the biology of all modules, running a GO term analysis on each module.

-What is the relationship between GOterm analysis and AUC? Don't you usually get a P-value as result?

-Paragraph 2.5: could you explain more on how the hypergeometric test is performed?line 153-154: In my opinion the P-value cutoff should be more stringent.

Results:

-Figure 1 can be graphically improved and still not sure about AUC calculation for GO term analysis, please explain 

-Would be nice to check if those 9 genes are significantly associated to recurrence. To support your results would be good to add a supplementary table describing the association for each gene in the module. This can be use as additional layer of filtering .

-Line 192-193: wrong sign on P-value and node degree.

-Improve quality of figures

-It's not clear how the RF is performed, evaluation and test of the RF model are missing. Without this, publication is not recommended
